# Transcriptomic and Metabolomic Analysis Provides Insights into the Fruit Quality and Yield Improvement in Tomato under Soilless Substrate-Based Cultivation

Jinghua Guo [1], Lingdi Dong [1,*], Shyam L. Kandel [2], Yonggang Jiao [1], Linqi Shi [1], Yubo Yang [1], Ainong Shi [3] and Beiquan Mou [4]

[1] Institute of Cash Crops, Hebei Academy of Agriculture and Forestry Sciences, Shijiazhuang 050051, China; 15831138918@163.com (J.G.); 15831193233@163.com (Y.J.); shilinqi113@163.com (L.S.); y15930880226@163.com (Y.Y.)
[2] Sugarbeet and Potato Research Unit, USDA-ARS, Edward T. Schafer Agricultural Research Center, Fargo, ND 58102, USA; shyam.kandel@usda.gov
[3] Department of Horticulture, University of Arkansas, Fayetteville, AR 72701, USA; ashi@uark.edu
[4] Crop Improvement and Protection Research Unit, USDA-ARS, Salinas, CA 93905, USA; beiquan.mou@usda.gov
* Correspondence: donglingdi@163.com

**Abstract:** The effects of soilless substrate-based versus soil cultivation on overall fruit quality and yield in tomato (*Solanum lycopersicum*) were studied using the tomato cv. Zhonghua Lvbao. Experiments for tomato soilless cultivation were carried out under greenhouse conditions. Plant growth, fruit quality and yield, and physiologic traits were observed. RNA-seq and RT-PCR, as well as metabolomic analyses were performed to examine the expressed genes and metabolites under soilless substrate cultivation. The results showed that the plant height, stem diameter, and chlorophyll contents of tomato under substrate-based cultivation were increased by 37.3%, 19.8%, and 15.3%, respectively, compared with soil cultivation system. Leaf photosynthetic and transpiration rates, stomatal conductance, and root vitality of tomato, under substrate-based cultivation, increased by 29.0%, 21.2%, 43.9%, and 84.5%, respectively, compared with soil cultivation. The yield reached 7177.5 kg/667 m$^2$, and the relative yield increased by 10.1%, compared with soil cultivation. The contents of total soluble sugar, soluble solids, and vitamin C increased by 35.7%, 19.7%, and 18.2%, respectively, higher than those of soil cultivation in tomato fruits, while nitrate content and titratable acid decreased by 29.4% and 11.8%, respectively. Therefore, substrate-based-cultivation can increase production and improve tomato fruit quality and taste. We examined the expressed genes and metabolites to explore the molecular mechanism of plant growth and overall fruit quality improvement in substrate-based cultivation. A total of 476 differentially expressed genes were identified by transcriptomes profiling, of which 321 and 155 were significantly up- and down-regulated, respectively. The results of metabolomics analysis showed that 441 metabolites were detected, where 24 and 36 metabolites were up- and down-regulated, respectively. By combining analyses of transcriptomic and metabolic groups, genes and metabolites related to the fruit quality were mainly concentrated in the vitamin B6/ascorbic acid/aldonic acidmetabolism, and glycerophospholipid metabolic pathways. Therefore, substrate-based cultivation can elevate vitamin and soluble sugar contents and the expression of fruit flavor related genes, which lays an initial background for exploring the mechanism of substrate-based cultivation, in order to improve the quality of tomato in the future.

**Keywords:** tomato; *Solanum lycopersicum*; metabolomics; transcriptomics; metabolic pathway; fruit quality

## 1. Introduction

Tomato (*Solanum lycopersicon* L.) is one of the major vegetables consumed worldwide and favored for its nutritional contents and unique flavor [1]. Consumption of tomato

provides people with a variety of natural vitamins, mineral elements, and antioxidant compounds [2]. Therefore, a higher nutritional value of tomato would be helpful in improving human health and wellness [3]. With the improved living standards and health awareness, people's demand for tomatoes gradually changes from quantity to quality, and they begin to pay more attention to the intrinsic quality of tomatoes [4]. The quality of the tomato is a comprehensive index, including flavor and nutritional quality, commodity quality, etc. The flavor quality consists of taste and volatile aroma and is related to vitamin C, lycopene, soluble solids, soluble sugar, amino acid contents, and sugar-to-acid mass ratio [5]. In recent years, tomato production is expanded significantly with higher inputs requiring higher fertilizers, water, pesticides, etc. Use of such tomato production system often leads to the negative consequences in soil health, increases in diseases and insect pests, results in poor yield and quality, and impacts the sustainable production of tomato and viability of the tomato industry [6,7].

Growing tomatoes in a soilless substrate can provide a viable alternative to avoid the problems of soil salinization and soil-borne diseases caused by long-term continuous cropping in traditional cultivation system [8]. Substrate-based cultivation is one of the main forms of soilless cultivation, which is not restricted by geographical regions, saves water and fertilizer resources, and possibly utilizes agricultural wastes [9]. At present, substrate-based technology has been widely used in more than 100 countries, and more than 90% of commercial soilless cultivation is substrate-based cultivation system [10]. Substrate-based cultivation system represents the development trend of modern facilities in agriculture [10]. Studies have shown that the substrate-based cultivation can provide good physical and chemical environments in the root systems and higher accessibility to the mineral nutrients and organic matter needed for tomato growth, thereby promoting growth, improving photosynthetic characteristics, and increasing yield and fruit quality [11,12]. Cai et al. [13] suggested that the improvement of tomato yield and fruit quality, under substrate-based cultivation, is largely associated with the ratio and total amount of available nitrogen, phosphorus, and potassium in the substrate and crop requirements. Sunera et al. [14] found that bacteria from the tomato rhizosphere significantly increased plant growth and the mineral uptake of plants. Studies have shown that compared with soil cultivation, substrate-based cultivation can significantly increase the yield, sugar and soluble solid contents, and sugar-to-acid ratio of the tomato fruit, as well as significantly reduce the nitrate content, thus improving tomato taste and quality [15]. Wang et al. [16] suggested that substrate-based cultivation could significantly increase the vitamin C contents, as well as increase the yield by 12.5–16.8% [16]. However, there are few reports on how substrate-based cultivation can improve tomato yield and quality at the molecular level.

To understand the molecular basis of the growth improvement in tomato under soilless cultivation, the research group used a self-developed, resource-saving substrate to study its effect on tomato yield, fruit quality, and taste in this study. Based on the combination of indoor experiments and quality tests, we utilized targeted metabolomics and transcriptomics approaches to understand the effects of two different cultivation systems, including substrate-based and conventional cultivation using soil. For the metabolomics study, we used liquid chromatography, coupled with the tandem mass spectrometry (LC-MS/MS) and Illumina platforms (Illumina, Inc., San Diego, CA, USA) for transcriptional analysis to identify candidate genes active under soilless cultivation, as well as functional annotation and enrichment analysis on them. The combined transcriptional and targeted metabolome analysis revealed the pattern of gene expression related to the key metabolic pathways that affect the quality of tomato fruit in substrate-based cultivation. This study provides new insights into improving the nutritional contents of tomato fruits and potentially efficient tomato production using soilless substrate-based cultivation.

## 2. Materials and Methods

### 2.1. Plant Materials and Production System

The tomato cultivar, 'Zhonghua Lvbao', released by Shijiazhuang Agricultural Doctor Technology Development Co., Ltd., Shijiazhuang, China, was used in this study under soilless substrate- and soil-based cultivation. The soilless substrate is comprised of cotton-seed skin residue, vermiculite, and perlite (1:1:0.2). The substrate was developed by the Economic Crop Research Institute of Hebei Academy of Agriculture and Forestry Sciences, Shijiazhuang, China.

The soil nutrient content and physical and chemical properties of soilless substrate vs. soil growth medium was listed in Table 1. The loamy soil was collected and transferred from the experimental field into greenhouse at the Modern Agricultural Experiment Park of the Hebei Academy of Agriculture and Forestry Sciences, Hubei province in China. We used the water-soluble fertilizer (N + $P_2O_5$ + $K_2O$ ≥ 52%, Mn + Zn + B ≥ 0.5%) produced by Hebei Mengbang Water-soluble Fertilizer Co., Ltd., Zhaoxian, China under soilless cultivation.

**Table 1.** Nutritional status, pH, and electrical conductivity of growth medium.

| Cultivation | pH Value | Nitrogen, Nitrate (mg/kg) | Phosphorus (mg/kg) | Potassium (mg/kg) | EC (uS/cm) |
|---|---|---|---|---|---|
| Soilless substrate | 7.1 | 378.25 | 320.2 | 500 | 470 |
| Soil | 7.66 | 213.12 | 378.5 | 250 | 473 |

### 2.2. Experimental Design

Experiments for tomato soilless cultivation were carried out for three consecutive years, from 2018 to 2021. For the transcriptomic study, the tomato cultivar, "Zhonghua Lvbao", was planted on 25 January 2019 in the soilless substrate-based cultivation, as well as on 15 March 2019 in the soil. The fertilizer was dissolved in water and applied through irrigation daily, as listed in Table 2. The agronomic practices were the same for both treatments.

**Table 2.** Fertilizer and irrigation scheme in the study.

| Substrate-Based Cultivation | Fertilizer Type | Amount of Fertilizer (kg/667 m²/day) | Amount of Water (1000 Liters/Day) | Day |
|---|---|---|---|---|
| Early fruit set stage (1–2 fruits) | Balanced water-soluble fertilizer 20-20-20N/K = 1:1 | 1~1.2 | 1~1.2 | 20 |
| Fruit set period (3–6 fruits) | High potassium water soluble fertilizer 15-5-30 N/K = 1:2 | 1.2~1.5 | 1.2~1.5 | 30~40 |
| | Middle element water-soluble fertilizer | 0.5 | | |
| Harvesting stage | no fertilizer | | 1~1.2 | 10 |

### 2.3. Measurement of Biological and Physiological Growth Parameters

Plant height, stem thickness, root volume, roots, stems, leaf fresh weight, and relative chlorophyll content were measured 50 days after planting. Fifteen plants were evaluated for each trait with three replicates. The chlorophyll content was measured with a 502-chlorophyll SPAD meter (Konica Minolta, Tokyo, Japan). Each leaf was measured three times in same day. The averages of the three readings was recorded and 15 plants were measured for each treatment. The root vigor and vitality of tomato in representative plants were taken at the later stage of growth and determined by using the triphenyltetrazolium

chloride reduction (TTC) method [17]. Measurement of photosynthetic characteristics of tomato, including photosynthetic rate (Pn), stomatal conductance (Gs), intercellular $CO_2$ concentration (Ci), and transpiration rate (Tr), were made by a TPS-2 portable photosynthesis meter (PP Systems, Amesbury, MA, USA.) 60 days after planting. The fourth leaf from the top, with similar light exposure on each plant, was selected to measure the photosynthetic characteristics. The measurement was conducted at 9:00-11:00 am on a sunny day once at the same time. The mature fruits were harvested several times when the fruit color turns completely red. The total numbers of fruits were not recorded from each treatment, but the yield was recorded, cumulatively harvesting in 50 plants growing in a 17-m long row and converted into yield per mu (1 mu = 667 $m^2$).

### 2.4. Tomato Fruit Quality Measurement

The tomato quality index was determined during the fruit harvesting time. The colorimetric method was used for the determination of nitrate [18]. The soluble sugar was determined by the anthrone colorimetric method [18]. Vitamin C (Vc) was determined by the 2,6-dichlorophenol indophenol titration method [18]. The titratable acid was determined by the indicator titration method [19], and the soluble solid content was determined by a handheld pocket refractometer, PAL-1 (ATAGO, Tokyo, Japan).

### 2.5. Differential Gene and Metabolites Screening

The tomato fruits were sampled three times for each treatment with fully ripe tomatoes when the fruit color turned completely red. The harvested fruits were quickly chopped into smaller pieces, transferred into test tubes, and immediately placed at −80 °C, until further processing. For transcriptomic study, total RNA was extracted using a quick RNA isolation kit (Huayueyang Biotech Co., Ltd., Beijing, China), according to the manufactures' instructions. The cDNA libraries were constructed and sequenced using the Illumina platform (Illumina, Inc., San Diego, CA, USA). Total RNA-seq reads were retrieved, and adapters were removed from all reads prior to analysis. Clean reads were used to map over the tomato reference genome, using HISAT2 (*Solanum lycopersicum* version number: SL3.0 (https://www.ncbi.nlm.nih.gov/genome/?term=txid4081, accessed on 6 April 2022). By using FPKM (fragments per kilobase of exon model per million mapped reads) values differentially expressed genes were identified by using DESeq software [20]. Genes expressed with fold change ≥ 2 and FDR (false discovery rate) *p*-value < 0.01 were used for further analysis. KOBAS [21] software was used to test the enrichment of significantly differentially expressed genes in the KEGG pathway. Ultra-high performance liquid chromatography and tandem mass spectrometry (UHPLC-MS) were used for analyzing metabolomics of tomato fruits. The orthogonal projections to latent structures discriminant analysis (OPLS-DA) was performed to preliminarily screen out metabolites that differ among different samples. The metabolites were further screened by combining the *p*-value or fold change of the univariate analysis. The following criteria were used to identify metabolomics significant to the substrate and soil cultivation. (1) Metabolites abundant with fold change ≥ 2 were selected. The difference between the control and experimental groups was considered to be significant if the difference in metabolites was more than two times. (2) If fold change value was duplicated between samples, metabolites with variable importance in projection (VIP) ≥1 were selected. The VIP value represents the influence strength of the corresponding metabolite between groups in the classification and discrimination of each group of samples in the model, and it is generally considered that the metabolites with VIP ≥ 1 are significantly different. Based on the platform LC-MS/MS, fold change ≥ 2 or ≤ 0.5, and VIP ≥ 1 criteria were used to screen for significantly different metabolites. The KEGG database [22] was used to annotate the differentially expressed metabolites. The combined analysis of transcriptomics and metabolomics studies was adopted to select metabolic pathways related to changes in tomato quality and find candidate genes related to tomato quality metabolism.

### 2.6. Quantitative PCR Verification

Six target genes (LOC101262853, LOC101268829, LOC543815, LOC101261156, LOC101262682, and LOC101246511) were included for quantitative reverse transcription-PCR (qRT-qPCR) assay in order to verify the expressed genes in the transcriptomics data. The qRT-PCR primer information is listed in Table 3, where 18S-F and 18S-R are primers from the 18S rRNA gene which was used as the control.

**Table 3.** qPCR primer information.

| Gene ID | Primer Name | Primer Sequence (5′ to 3′) | Tm (°C) |
|---|---|---|---|
| 1 | 18S-F | CAACCATAAACGATGCCGA | |
| | 18S-R | AGCCTTGCGACCATACTCC | |
| 2 | 101268829-F | CACTGTCTTCATCATCATCTTCTG | |
| | 101268829-R | GTAATCGCCTCGGATCTTCT | |
| 3 | 101261156-F | GATCACCTCCTGGTTGCCTTAG | |
| | 101261156-R | AAGAAGTTGGATTTGCCCTGAAGA | |
| 4 | 101262682-F | ACTGGTAGATCGTCTGCTA | 60 |
| | 101262682-R | CACATTCCTCATCGGTCAA | |
| 5 | 101246511-F | GATGCTCTTTGCCAGTTTGTG | |
| | 101246511-R | GCCAACTCCTTCATGTCCAA | |
| 6 | 543815-F | CACTCCTCCCTGAAGATCCTTAT | |
| | 543815-R | TTGCTCCACTCCGACCTT | |
| 7 | 101262853-F | CCAGATACTGTGCCAATCAGAACT | |
| | 101262853-R | ACCAATGCCTATGTCGTAGAATCC | |

The qRT-PCR reaction components included 500 ng RNA, 4 μL 5 × RT reaction mix, 0.8 μL Rondam primer/oligodT, 0.8 μL TUREscript—RTase/RI mix, and addition of RNase-free dH$_2$O to total volume of 20 μL. The qRT-PCR assay involved: Step 1: 95 °C-3 min., one cycle; Step 2: 95 °C-10 s, and 60 °C-30 s, 39 cycles, holding at 1 °C. The cDNA synthesized from qRT-PCR assay was stored at −80 °C, until further processing.

### 3. Results

#### 3.1. The Effects of Different Treatments on Tomato Growth, Yield and Quality

Physical plant growth parameters including plant height, stem thickness, chlorophyll content, fresh root/stem/leaf weight, and root volume under soilless substrate-based cultivation were significantly higher than in the soil cultivation, which were increased by 37.3%, 19.8%, 15.3%, 151.7%, 53.3%, 116.9%, and 283.4%, respectively, compared to the plants under soil cultivation (Table 4). The tomato plants grown under substrate-based cultivation showed stronger growth vigor (Figure 1), a bigger root system (Figure 2), and increased fruit yield (by 10.1%) under substrate-based cultivation than under soil cultivation.

**Table 4.** Tomato growth and yield under substrate and soil cultivation conditions.

| Cultivation | Plant Height (cm) | Stem Width (mm) | Fresh Root Weight (kg) | Fresh Stem Weight (kg) | Leaf Weight (kg) | Root Volume (mL) | Yield (kg/667 m$^2$) | Chlorophyll Content (%) |
|---|---|---|---|---|---|---|---|---|
| Substrate | 100.07 a | 8.71 a | 0.073 a | 0.46 a | 1.28 a | 76.67 a | 7177.50 a | 52.25 a |
| Soil | 72.87 b | 7.27 b | 0.029 b | 0.30 b | 0.59 b | 20.00 b | 6517.99 b | 45.33 b |
| Difference | 27.2 | 1.44 | 0.044 | 0.16 | 0.69 | 56.67 | 659.51 | 6.92 |
| Increase% | 37.3 | 19.8 | 151.7 | 53.3 | 116.9 | 283.4 | 10.1 | 15.3 |

Difference = substrate cultivation − soil cultivation. Increase% = 100 × difference/soil cultivation. The significant difference (a, b) is at $p$-value = 0.05 level.

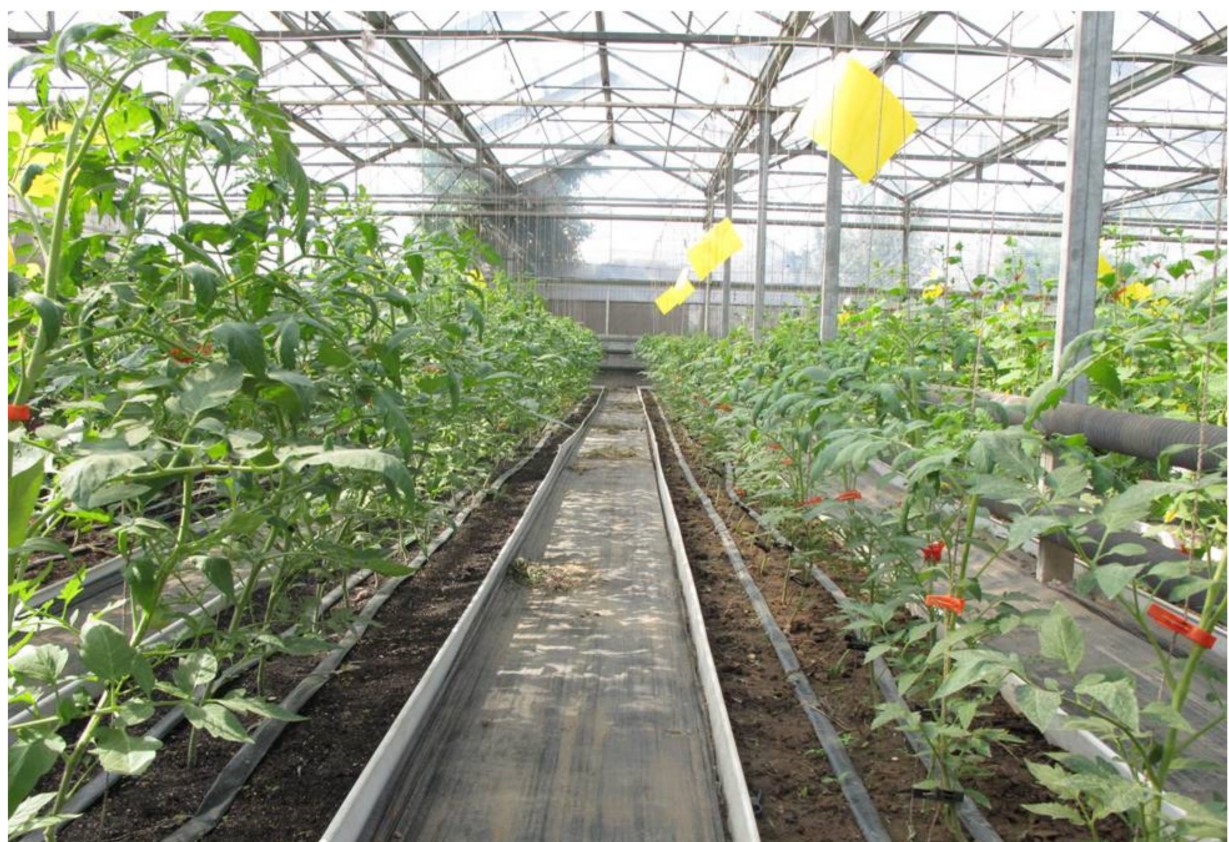

**Figure 1.** Visual growth difference in representative tomato plants under substrate-based (**left**) and soil cultivation (**right**).

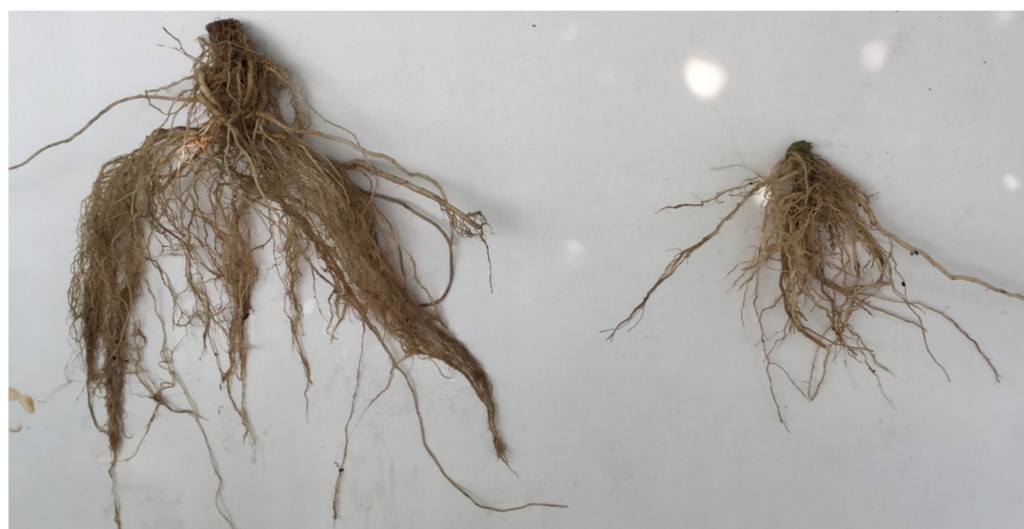

**Figure 2.** Root growth of representative tomato plant between substrate-based (**left**) and soil cultivation (**right**).

The photosynthetic and transpiration rates, stomatal conductance, and root vitality of tomato grown in soilless substrates were significantly higher and increased by 29.0%, 21.2%, 43.9%, and 84.5%, respectively, compared with soil cultivation. The intercellular $CO_2$ concentration was decreased by 2.3%; however, the difference was not statistically significant (Table 5). Photosynthesis is an important physiological process in plants, which directly influences the plant growth, yield, and host immunity. Photosynthesis can be used

as an important indicator to evaluate plant growth. Photosynthetic rate was significantly higher in tomato plants grown under soilless substrate-based cultivation than conventional soil cultivation (Table 5).

**Table 5.** Photosynthetic characteristics and root activity of tomato under substrate and soil cultivation conditions.

| Cultivation | Photosynthetic Rate ($\mu$mol/m$^2$/s) | Transpiration Rate (mmol/m$^2$/s) | Stomatal Conductance (mol/m$^2$/s) | Root Vitality $\mu$g$\cdot$g$^{-1}\cdot$h$^{-1}$ | Intercellular CO$_2$ Concentration ($\mu$mol/mol) |
|---|---|---|---|---|---|
| Substrate | 15.05 a | 7.42 a | 0.59 a | 373.24 a | 300.78 a |
| Soil | 11.67 b | 6.12 b | 0.41 b | 202.34 b | 307.78 a |
| Difference | 3.38 | 1.3 | 0.18 | 170.9 | −7 |
| Increase% | 29.0 | 21.2 | 43.9 | 84.5 | −2.3 |

Difference = substrate cultivation − soil-based cultivation. Increase% = 100 × difference/soil-based. The significant difference (a, b) is at *p*-value = 0.05 level.

The nutritional components, such as total soluble sugar, soluble solids, and vitamin C contents were increased by 35.7%, 19.7%, and 18.2%, respectively under soilless substrate-based cultivation (Table 6), indicating that the substrate-based increased the nutritional values of the tomato. However, undesirable components for fruit quality, such as concentrations of nitrate and titratable acid were significantly decreased by 29.4% and 11.8% in substrate-based cultivation (Table 6). Nitrate is an important indicator to measure the quality of vegetables with negative impact [23]. A total of 70–80% of the nitrate ingested by humans comes from vegetables. When it accumulates in the body to a certain level, nitrate may affect the human body, causing serious health hazards [24]. Excessive nitrate content in the tomato fruit is considered a detrimental factor for human health [23,24]. Soluble sugar and organic acid contents are important indicators for the favorable taste of tomatoes. A proper sugar-to-acid ratio can produce fruits with good flavor. In our experiments, the sugar-to-acid ratio significantly improved (increased by 53.64%) under substrate-based cultivation, which can help to improve the flavor of tomatoes.

**Table 6.** Tomato quality under substrate-based and soil cultivations.

| Cultivation | Total Soluble Sugar Content (%) | Soluble Solid Content (%) | Vitamin C (mg/100 g) | Sugar-to-Acid Ratio | Nitrate (mg/kg) | Titratable Acid (%) |
|---|---|---|---|---|---|---|
| Substrate cultivation | 40.84 a | 5.23 a | 28.33 a | 9.08 a | 202.99 b | 0.45 b |
| Soil cultivation | 30.09 b | 4.37 b | 23.97 b | 5.91 b | 287.45 a | 0.51 a |
| Difference | 10.75 | 0.86 | 4.36 | 3.17 | −84.46 | −0.06 |
| Increase% | 35.7 | 19.7 | 18.2 | 53.6 | −29.4 | −11.8 |

Difference = substrate cultivation − soil-based cultivation. Increase% = 100 × difference/soil-based. The significant difference (a, b) is at *p*-value = 0.05 level.

*3.2. Transcriptional Analysis of Tomato Fruits under Substrate-Based and Soil Cultivation Conditions*

3.2.1. Pre-Processing of Transcriptomics Data

A total of six tomato RNA samples from substrate and soil cultivation were subjected to high-throughput sequencing on the Illumina platform. A total of 49.94 GB clean RNA-seq data was obtained from Illumina sequencing platform. The clean reads of each sample were mapped into the tomato reference genome. Reads of each sample mapped with the tomato reference genome were between 95.5% and 97.1%. From mapping of RNA-seq reads to the tomato reference genome, 1383 differentially expressed genes were discovered, of which, 945 genes were functionally annotated.

### 3.2.2. Identification of Differentially Expressed Genes in Tomato Fruits under Substrate-Based and Soil Cultivation Conditions

A total of 476 significantly differentially expressed genes were identified between the two cultivation systems (Figure 3), among which 321 and 155 genes were significantly up- and down-regulated, respectively (Supplementary Table S1).

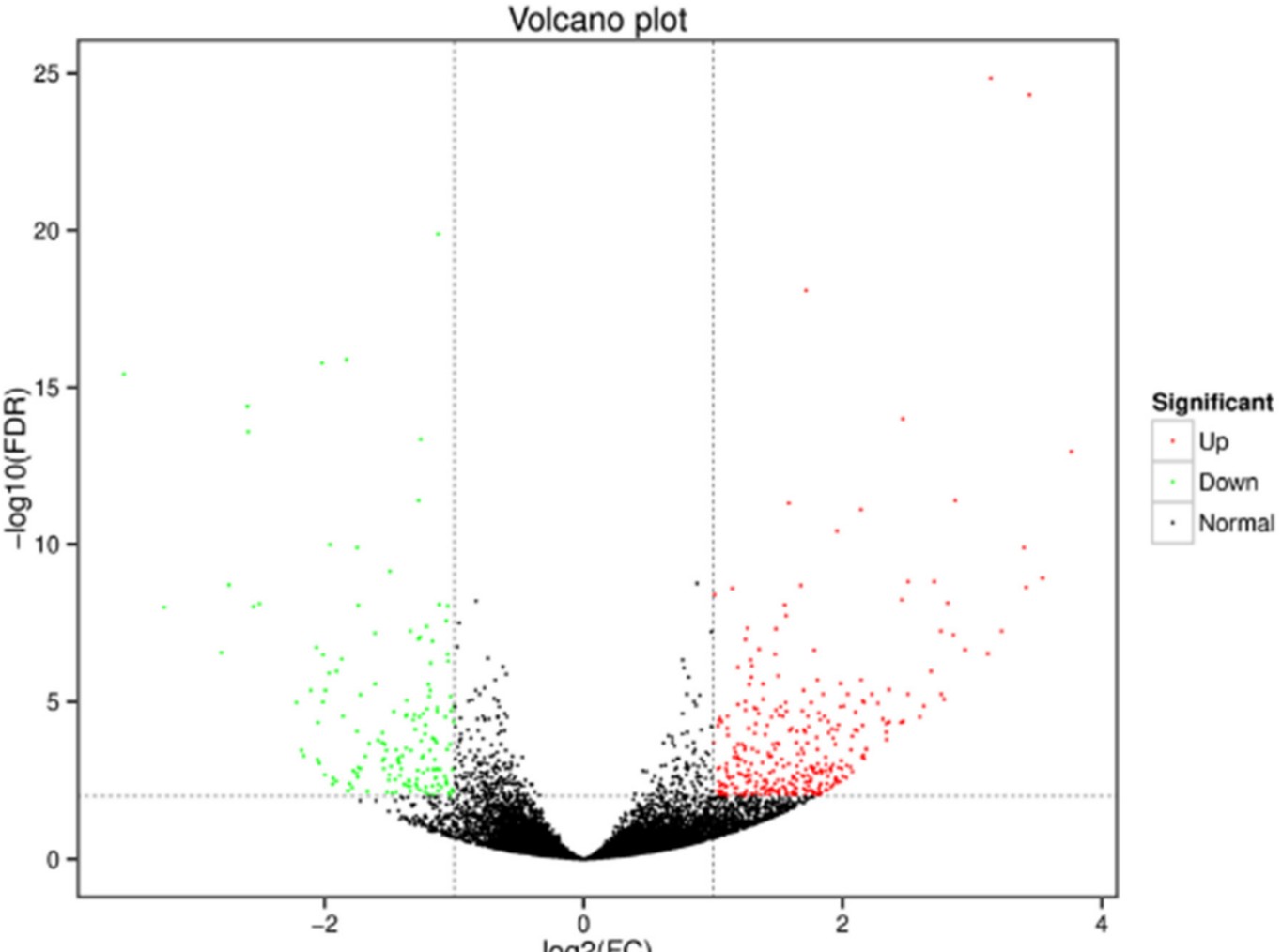

**Figure 3.** Volcano plot of differential expression. Green dots represent significantly down-regulated genes, red dots represent significantly up-regulated genes, and black dots represent insignificant differential expression.

### 3.2.3. Functional Annotation of Significantly Expressed Genes

Gene ontology (GO) classification analysis was performed using significantly differentially expressed genes, where 362 out of 476 significantly differentially expressed genes were annotated (Figure 4). Significantly, expressed genes enriched in the biological process were mainly related to the classification of metabolic/cellular/single organism processes and stimulus responses. Expressed genes enriched in cellular components were mainly related to cell parts, organelles, and membranes; enriched genes in molecular functions were mainly associated with polymerization, catalytic, and nucleotide transcription factor activities, etc.

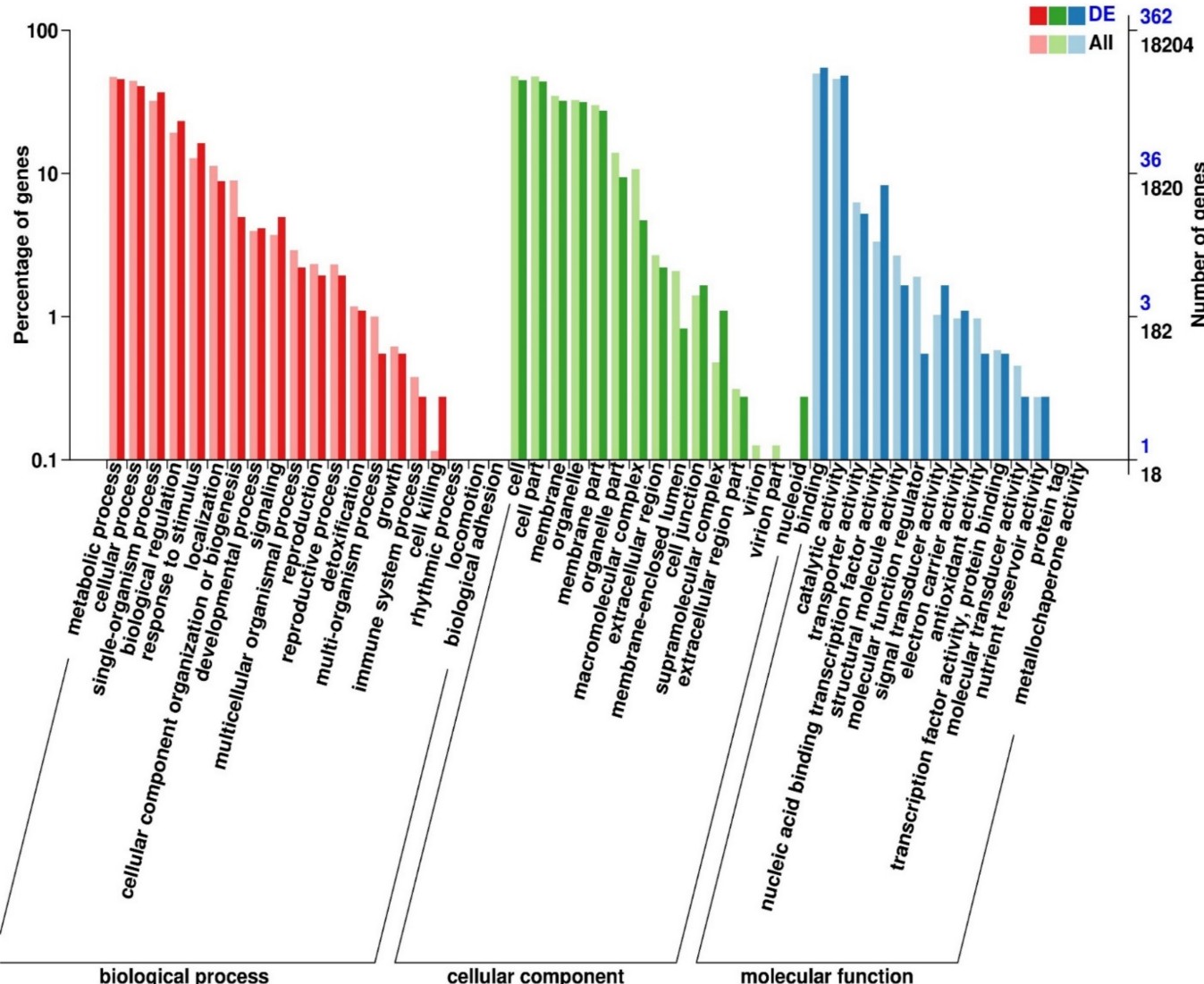

**Figure 4.** GO annotation classification of significantly differentially expressed genes (Log$_2$fold change $\geq$ 2 and FDR *p*-value $\leq$ 0.01).

KEGG pathway enrichment results showed that 476 significantly differentially expressed genes were enriched in 71 pathways. The main pathway enrichment included carbon metabolism, sucrose, and starch metabolism, vitamin B6/ascorbic acid/aldonic acid/glycerophospholipid/amino acid/ribonucleotide metabolism, amino acid biosynthesis, linoleic acid metabolism, and other pathways (Figure 5).

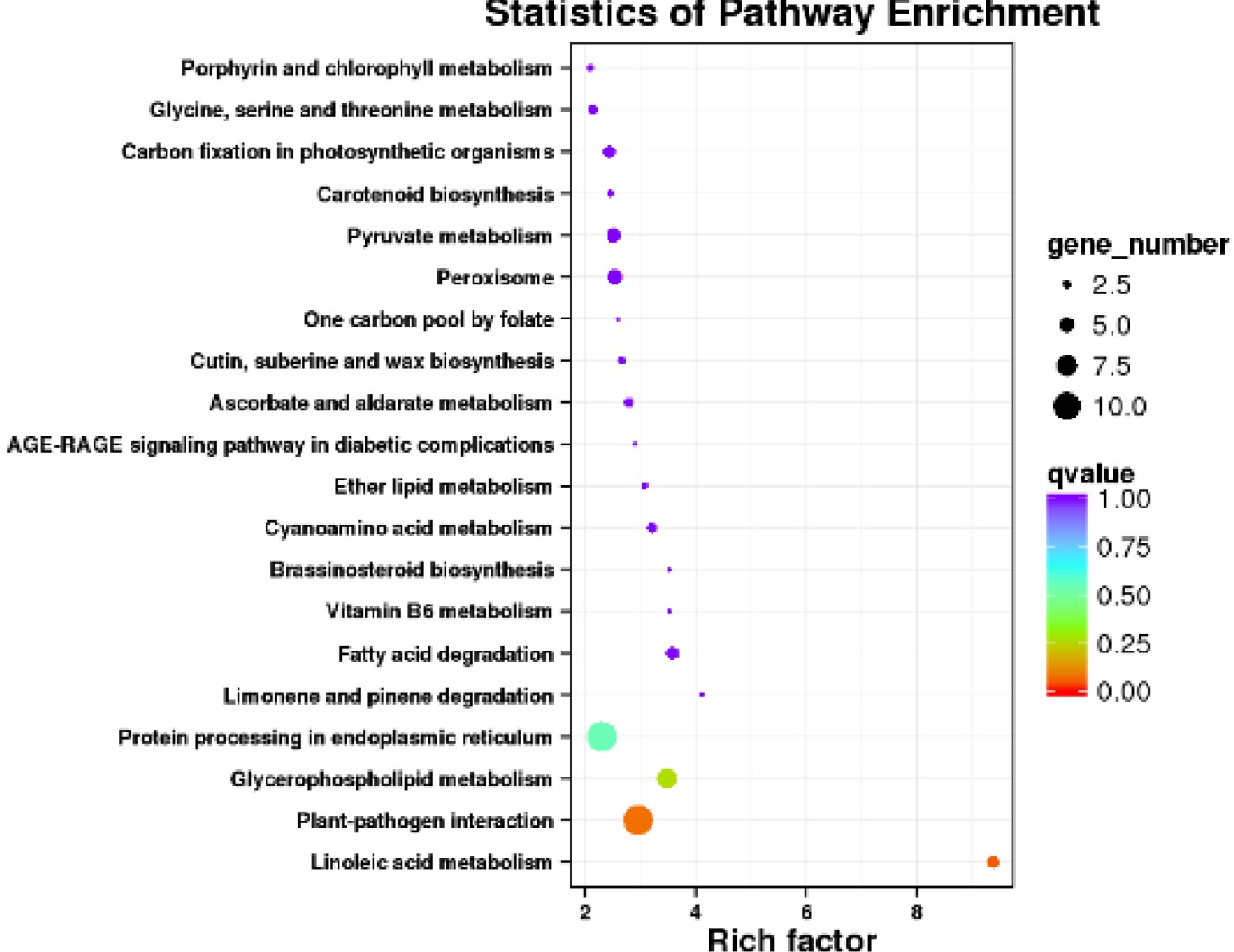

**Figure 5.** KEGG pathway enrichment of significantly differentially expressed genes (Log2fold change ≥ 2 and FDR *p*-value ≤ 0.01).

### 3.2.4. qRT-PCR Assay for Verification of Differentially Expressed Genes

To verify the expression status of RNA-seq results, six differentially expressed genes in the two treatment groups were selected for qRT-PCR analysis (Figure 6). Six genes had much higher expression in substrate-based than soil cultivation. We confirmed that the expression status of the six transcripts used in the qRT-PCR assay were consistent with the results of the RNA-seq analysis.

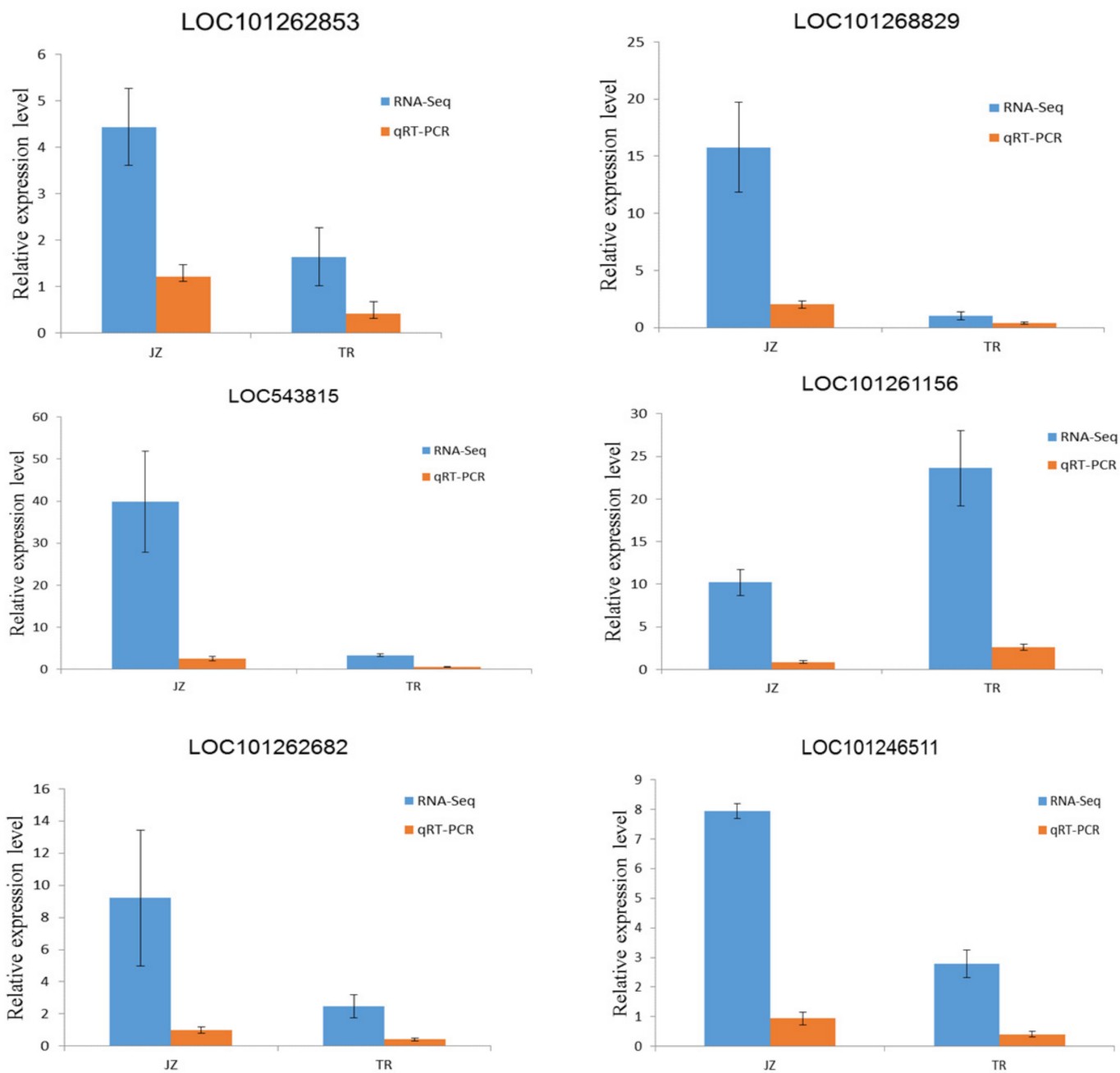

**Figure 6.** qRT-PCR verification results. JZ = Soilless substrate. TR = Soil.

*3.3. Screening Results of Different Metabolites in Tomato Fruits under Different Cultivation Conditions*
3.3.1. Principal Component Analysis of Differential Metabolites

Principal component analysis (PCA) can be used to examine the variability as magnitude of differences between the samples and within or across the treatments. From the PCA analysis (Figure 7), we found that PC1 (43.75%) separates metabolite contents between treatments, indicating that the metabolites of tomato fruits under substrate-based and soil cultivation are significantly different at *p*-value = 0.01.

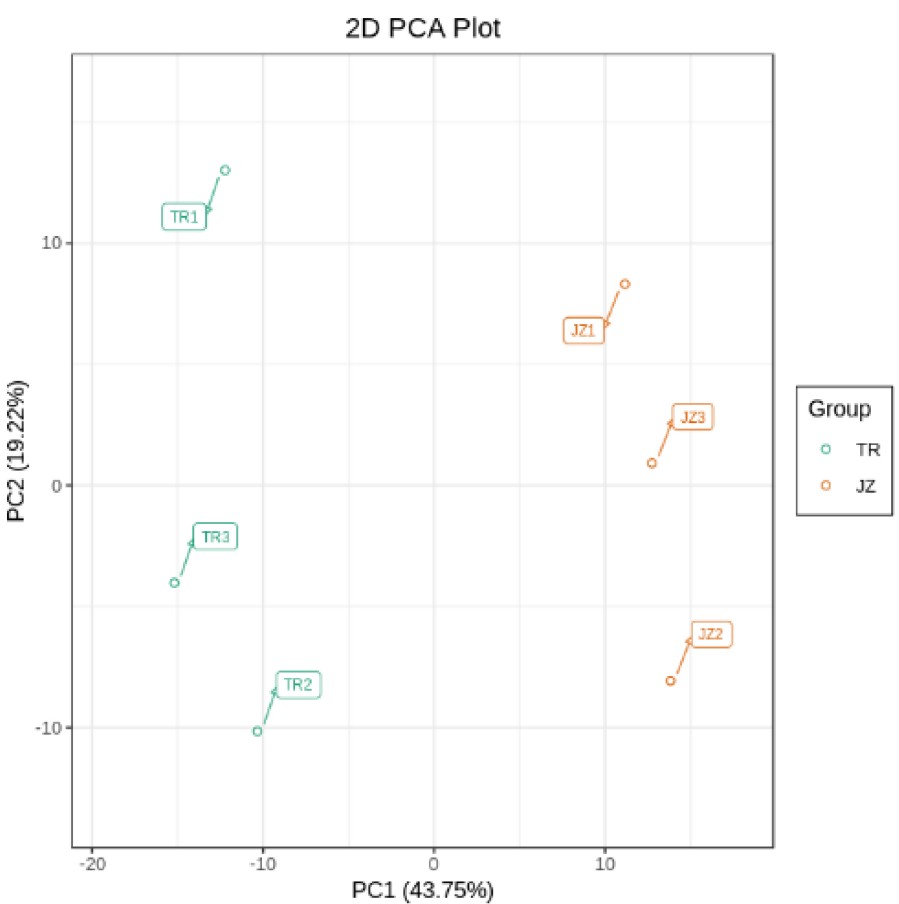

**Figure 7.** The principal component analysis of tomato fruit metabolites under different cultivation conditions.

### 3.3.2. Identification of Differentially Available Metabolites in Tomato Fruits

By using the ultra-high performance liquid chromatography tandem mass spectrometry (UHPLC-MS) detection platform, tomato fruits grown under two cultivation systems were subjected to targeted metabolome detection. A total of 441 metabolites were detected, of which, 60 significantly differentially generated metabolites were identified: 24 metabolites were up-regulated and 36 metabolites were down-regulated (Supplementary Table S2). The up-regulated metabolites are organic acids (diethyl phosphate), vitamins (pyridoxine O-hexoside, 4-pyridoxic acid), lignans and coumarins (pinoresinol diglucoside, oleurin diglucoside, and lipid diglucoside), alkaloids (serotonin, p-coumaroyl feruloyl caffeoyl spermidine), lipids (lysophosphatidylcholine and lysophosphatidylethanolamine), nucleotides and their derivatives (uridine 5′-diphosphate), amino acids and their derivatives (S-allyl-L-cysteine), and flavonoids (glycitin). Among the top 20 metabolic pathways that showed significant difference, tomato fruit quality-related metabolites were mainly enriched in the vitamin B6 metabolic, glycerophospholipid metabolism, the tryptophan metabolic pathway, and pyrimidine metabolic pathways (Figure 8).

There were three metabolites annotated in the vitamin B6 metabolic pathway: 4-pyridoxine and pyridoxine O-hexosides were up-regulated, and pyridoxine was down-regulated accounting for 10.7% of the total number of metabolites annotated (Figure 9). Three metabolites were annotated to the pyrimidine metabolism pathway: uridine 5′-diphosphate was up-regulated, and β-pseudouridine and uridine were down-regulated, which accounts for 10.7% of the total number of annotated metabolites (Figure 9). Among the other three metabolites of the glycerophospholipid metabolism pathway, lysophosphatidylcholine (lysophosphatidylcholine 14:0 (2n isoform) was up-regulated, and β-pseudouridine and uridine were down-regulated, accounting for another 10.7% of the total number of annotated metabolites (Figure 9).

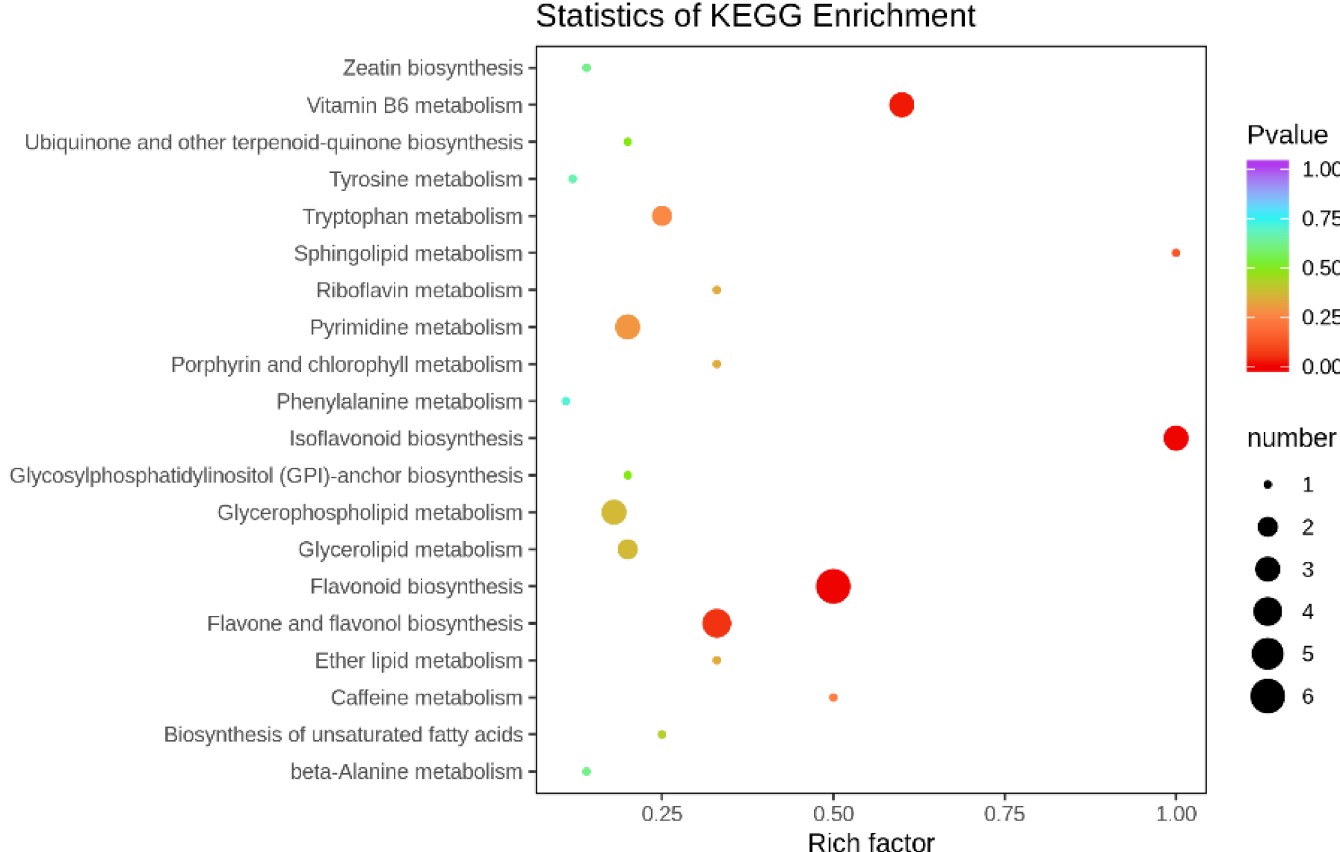

**Figure 8.** The KEEG enrichment of differentially available metabolites in tomato fruits.

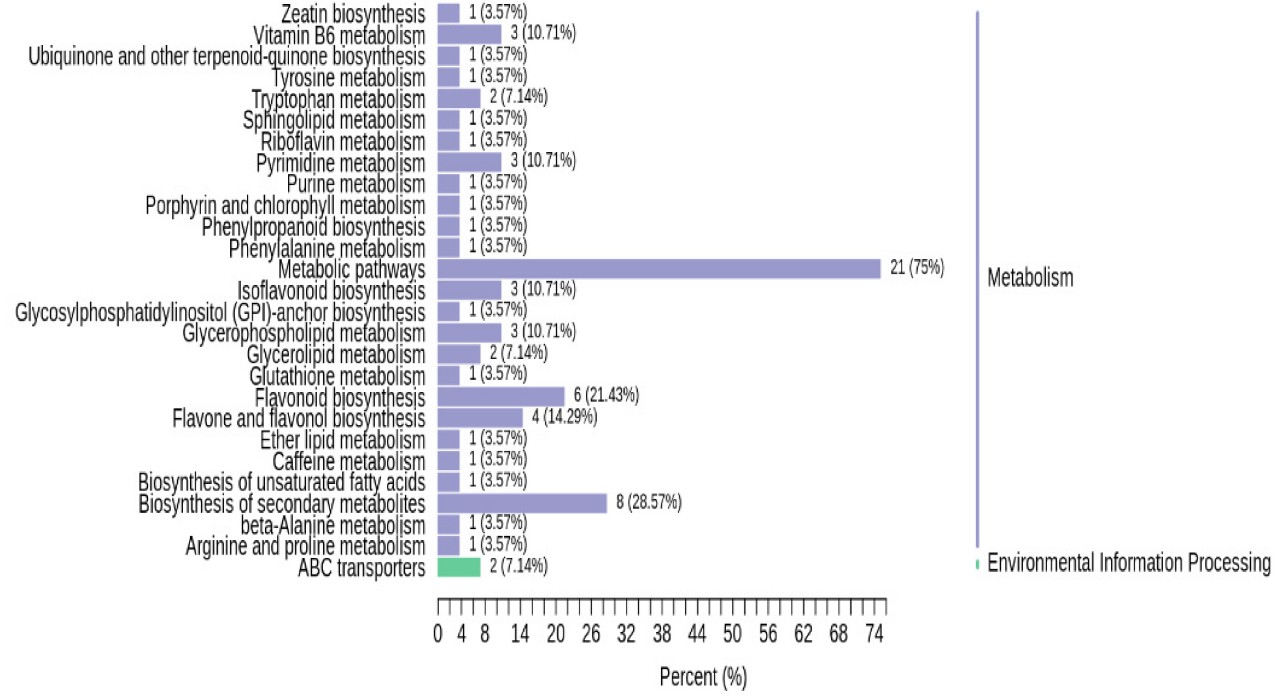

**Figure 9.** The KEGG classification of differential metabolites.

### 3.4. Combined Metabolomics and Transcriptomics Analysis

Based on the combined analysis of transcriptomics and metabolomics data (Figure 10), a total of 32 and 24 differentially expressed genes and metabolites, respectively, were enriched in 16 metabolic pathways (Supplementary Table S3). After combining the enrichment maps of the first 20 KEEG pathways, the quality-related expressed genes and metabolites were mainly enriched in the vitamin B6 metabolic, ascorbic acid aldehyde metabolism, and glycerophospholipid metabolism pathways.

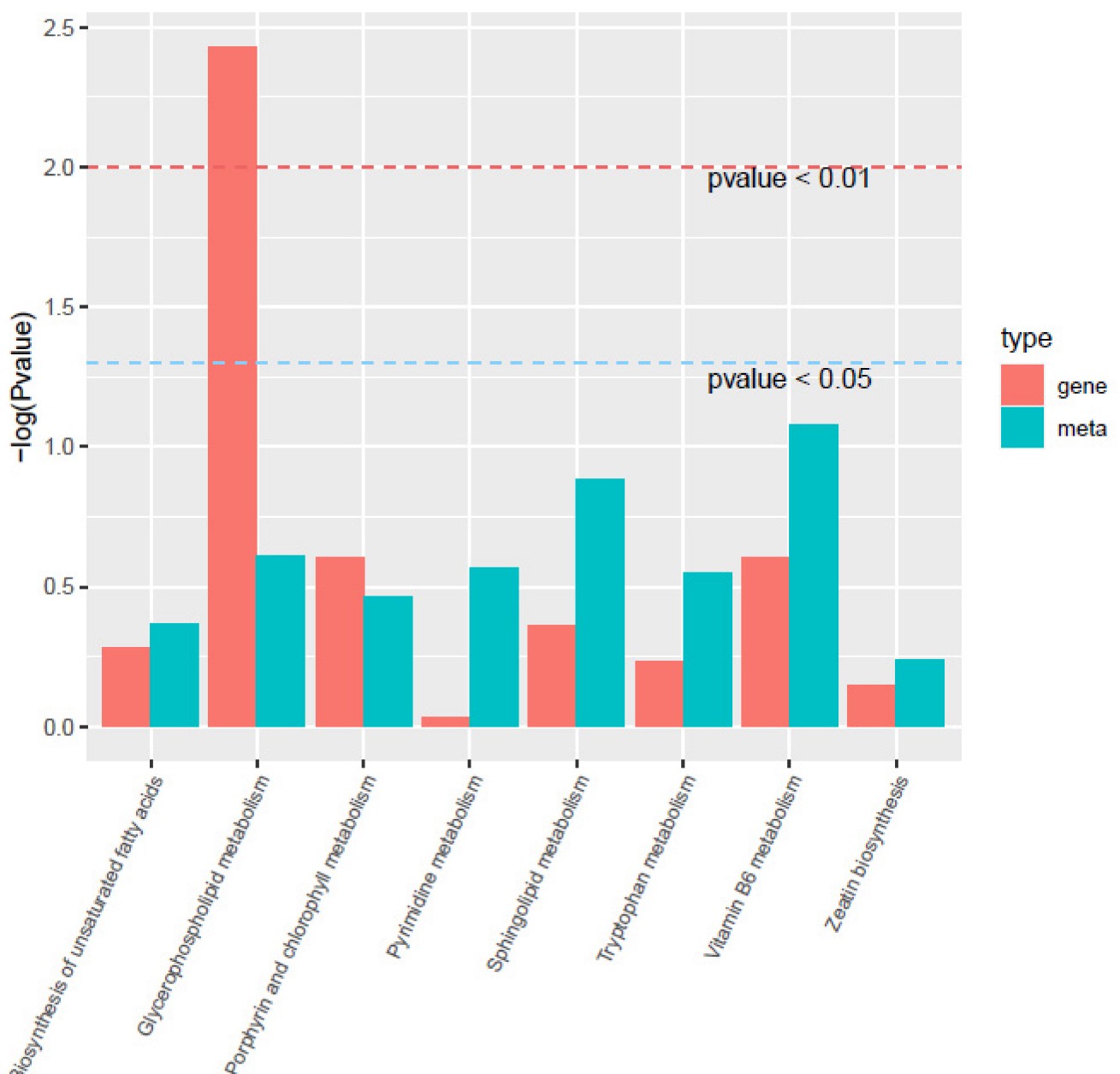

**Figure 10.** The KEGG enrichment of differentially expressed genes and metabolites.

### 4. Discussion

In this study, the substrate-based cultivation increased the relative content of chlorophyll, net photosynthetic rate, stomatal conductance, and transpiration rate of tomato leaves. Similar studies have shown that the combined application of bacterial residues and chemical fertilizers can significantly increase the chlorophyll content and photosynthetic rate of functional leaves during rice grain filling and delay the senescence of flag leaves

in the later stage, thereby increasing rice yield [25]. It is reported that the photosynthetic rate of crops is positively correlated with the nitrogen content of leaves [26,27]. In addition, potassium can regulate stomatal movement and play an important role in improving photosynthesis, promoting sugar accumulation, and increasing yield [28]. Zhang et al. [29] also showed that the application of potassium fertilizer could improve the nutritional quality of vegetables, increasing the Vc content of vegetables and reducing the nitrate content [29]. Higher potassium available in the substrate-based medium in this study also suggested the positive correlation between the higher potassium content and nutritional contents in tomato fruits. In addition, soluble sugar and organic acid contents are important indicators that affect the taste of tomatoes. A higher sugar-to-acid ratio is essential in tomato fruits to improve the taste and flavor. The results of this study showed that the sugar-to-acid ratio of substrate-based cultivation is 53.6% higher than that of soil-grown tomatoes. Therefore, substrate-based cultivation can improve the flavor and taste of tomatoes.

In this study, upregulated genes were significantly enriched in the linoleic acid metabolism pathway. The up-regulated gene TomloxE, which encodes for lipoxygenase (LOX) (EC: 1.13.11.58) enzyme, may promote the formation of flavor compounds in tomato fruits. Previous studies have shown that the activity of ethylene-regulated enzymes can be mainly regulated by the LOX gene family in fruits, among which TomloxA, TomloxB, and TomloxC are involved for the synthesis of fruit aromatic substances [30]. Furthermore, the LOX gene family also plays an important role in the formation of important agronomic traits, such as fruit aroma production [31–34] and postharvest storability [35,36]. Fatty acids, such as linoleic acid and linolenic acid are the main precursors in the formation of tomato fruit flavor through metabolism of fatty acids into volatile aromatic substances. This metabolic pathway is usually catalyzed by the LOX, lipid hydroperoxide lyase system, and oxidoreductase systems related catalytic reactions [37]. Therefore, the significant up-regulation of the lipoxygenase gene (EC: 1.13.11.58) in this study is perhaps involved in producing the tomato fruit flavor substances in the linoleic acid metabolism pathway. Transcriptomic analysis was also used to reveal genes tolerating manganese stress in *Schima superba*, a subtropical evergreen tree widely used for forest firebreaks and gardening [38].

Vitamin C is one of the essential micronutrients for humans [39]. In this study, an increased activity of uridine diphosphate glucuronidase (EC: 1.1.1.22) in the ascorbic acid and aldonic acid metabolism pathways promotes the formation of uridine diphosphate glucuronate, an important intermidate compound in the vitamin C biosynthesis pathway [40]. In this study, we found that substrate-based cultivation increased the soluble sugar and $V_C$ contents in tomatoes, which may be related to the up-regulation of genes LOC101246311 (UDP-glucose 6-dehydrogenase 1) and LOC101263958 (UDP-glucose 6-dehydrogenase 2). Phosphoethanolamine methyltransferase (PEAMT, EC2.1.1.103) belongs to a member of the nitrogen methyltransferase family and is a key enzyme in the biosynthetic pathway of plant phosphocholine. It can catalyze the reaction of phosphoethanolamine to receive the methyl group from S-adenosine methyl sulfide, which synthesizes phosphorylcholine [41–43]. Phosphatidylcholine is the most abundant phospholipid in the plasma membrane of plant cells, which facilitates the biological function of plasma membrane and improves the tolerance of plants in adverse environmental conditions [44,45]. In the glycerophospholipid metabolism pathway, the increase in the activity of glycerophosphodiesterase GDPD6 (EC: 3.1.4.46) promotes the increase in the content of phosphoethanolamine and activity of phosphoethanolamine N-methyltransferase (EC: 2.1.1.103), which may be related to the increased expression of genes LOC543681 (phosphoethanolamine N-methyltransferase) and LOC101261241 (glycerophosphodiester phosphodiesterase GDPD6) [46]. The human body requires four fat-soluble vitamins (vitamins A, D, E, and K) and nine water-soluble vitamins (vitamins B1, B2, B3, B5, B6, B7, B9, B12, and vitamin C). Vitamin B6 includes pyridoxal, pyridoxine, and pyridoxamine, as well as its phosphorylated derivatives. Prior studies have shown that the vitamin intake is still lower than the recommended level, indicating a significant gap between the human body's demand for vitamins and the actual intake [47]. At present, many bio-fortified foods can be eaten without cooking,

and vitamin bio-fortified food crops can have a positive impact on human nutrition and health [48]. This study shows that, in the vitamin B6 metabolic pathway, the increased activity of threonine synthase (EC: 4.2.3.1) promotes the increase in the content of vitamin pyridoxine O-hexosides and 4-pyridoxine, which may be related to the up-regulation of the gene LOC101260521 (threonine synthase). Prior studies have shown that amino acid metabolism is closely related to vitamin metabolism [48]. Fruit maturation is a physiological and biochemical process. In this process, the expression of intrinsic genes, for traits such as hardness, taste, aroma, and color of the fruit, can be adjusted to gradually change and achieve the purpose of regulation [49]. The upregulation of genes and accumulation of metabolites, related to the flavor, aroma, and taste of tomato fruits, were observed in plants grown under substrate-based cultivation. With the all-round development of transcriptomics, metabolomics, and whole genomics, new technical channels have been opened for the study of fruit quality regulation. The physical and chemical properties of substrate-based cultivation are important factors that affect the yield and quality of tomatoes. However, which components affect the key quality and flavor substances of the tomato fruit still need to be further studied.

## 5. Conclusions

The results of our experiments suggested that soilless substrate-based cultivation can improve overall plant growth, fruit yield, and quality better than soil-based conditions. Soilless substrates perhaps provided an optimum growth environment by improving nutrients acquisition and root aeration. The combined results of transcriptional and metabolomic analysis suggested that soilless substrate-based cultivation supported the expression and activation of genes and metabolomes related to the vitamin contents, as well as flavor and taste of tomato fruits. Future studies, leading to individual metabolic pathways and gene expression profiling, will help to understand the underlying mechanism of growth and quality improvement under substrate-based cultivation in detail.

**Supplementary Materials:** The following supporting information can be downloaded at: https://www.mdpi.com/article/10.3390/agronomy12040923/s1. Supplementary Table S1: Differentially expressed genes and their functional information. Supplementary Table S2: Differentially active metabolites and functional information. Supplementary Table S3: The KEGG enrichment of differential active metabolites.

**Author Contributions:** L.D. was the principal investigator (PI) of the project. J.G. collected and analyzed the experimental data. J.G., L.D., Y.J., L.S. and Y.Y. were involved in the phenotyping and performed metabolomics and transcriptomics experiments. J.G. wrote the first draft, and S.L.K., A.S., B.M. and L.D. edited the manuscript. All authors have read and agreed to the published version of the manuscript.

**Funding:** The research was supported by the key research and development plan project of Hebei Province in China "Research and Integration of Factors Affecting the Quality of Facility Tomatoes and Key Technologies of High-quality Cultivation", with the project number 20326903D.

**Institutional Review Board Statement:** Not applicable.

**Informed Consent Statement:** Not applicable.

**Data Availability Statement:** All data are included in tables, figures, and Supplementary Materials.

**Conflicts of Interest:** The authors declare no conflict of interest.

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
