# Peer review of "Transcriptomic and Metabolomic Analysis Provides Insights into the Fruit Quality and Yield Improvement in Tomato under Soilless Substrate-Based Cultivation"

_agronomy, doi:10.3390/agronomy12040923_

Round 1

Reviewer 1 Report

Overall, the study is well designed and presented in a good way, but mostly the literature is not cited i.e., in the 2nd line of introduction.

Abstract

The authors elaborated abstract in a good way. However, the specific methods are missing.

Introduction

The introduction part is well written, but several statements must be cited. Such as “Consumption of tomato provides people with a variety of natural vitamins, mineral elements, and antioxidant compounds”. Also enrich your articles with some recent studies in different sections like 10.1016/j.micpath.2020.103966; 10.3389/fgene.2021.635043.

Results and discussion

Results and discussion are well written and presented but the quality of figures is not good. Also cite some recent references as mentioned above in discussion sections.

Conclusion

Conclude your study in 1-2 paragraphs.

Author Response

Response to reviewer 1:

Overall, the study is well designed and presented in a good way, but mostly the literature is not cited i.e., in the 2nd line of introduction.

Response: Appreciate reviewer’s comment! We add several references.

Abstract

The authors elaborated abstract in a good way. However, the specific methods are missing.

Response: Excellent point! We add the methods in the abstract.

Introduction

The introduction part is well written, but several statements must be cited. Such as “Consumption of tomato provides people with a variety of natural vitamins, mineral elements, and antioxidant compounds”. Also enrich your articles with some recent studies in different sections like 10.1016/j.micpath.2020.103966; 10.3389/fgene.2021.635043.

Response: Excellent comments! Several more articles were cited.

Results and discussion

Results and discussion are well written and presented but the quality of figures is not good. Also cite some recent references as mentioned above in discussion sections.

Response: We will submit the quality of Figures and Tables in separate files.

Conclusion

Conclude your study in 1-2 paragraphs.

Response: Yes. We re-format the conclusion.

Reviewer 2 Report

Interesting research and has high potential value. I am concerned that I cannot determine what the soil medium is - what type of soil and soil structure as this can significantly impact tomato production. I also am not sure that the two different soilless media mentioned at the start of materials and methods would produce identical results. Vermiculite and perlite have quite different properties in soilless media. This needs to be better explained so that others can understand exactly what has been done so that results could be replicated. I have made some comments in the left side of paper and highlighted some areas that I think need rewording or that might be better explained.

Author Response

Response to Reviewer 2

Interesting research and has high potential value. I am concerned that I cannot determine what the soil medium is - what type of soil and soil structure as this can significantly impact tomato production. I also am not sure that the two different soilless media mentioned at the start of materials and methods would produce identical results. Vermiculite and perlite have quite different properties in soilless media. This needs to be better explained so that others can understand exactly what has been done so that results could be replicated. I have made some comments in the left side of paper and highlighted some areas that I think need rewording or that might be better explained.

Response: Appreciate reviewer’s comments and suggestions! We have made some changes based on the comments and suggestions.

Line 44, response: “and more” is deleted.

Lines 49-52, Sentence needs to be reworded to make good English sense. Maybe start with “"Use of such tomato production ...".

Response “Such tomato production system often leads to the negative consequences in soil health, increase in diseases and insect pests resulting poor yield and quality, and impacts the sustainable production of tomato and viability of the tomato industry”.

Changed to “Use of such tomato production system often leads to the negative consequences in soil health; increases in diseases and insect pests; results in poor yield and quality; and impacts the sustainable production of tomato and viability of the tomato industry [5-6].”.

Lines 53-55, Again, the start of sentence leads to imprecise English. Maybe start with "Growing tomatoes in a soilless substrate ...

Response: “Soilless substrate-based cultivation can provide a viable alternative to avoid the problems of soil salinization and soil-borne diseases caused by long-term continuous cropping in traditional cultivation system”

Changed to

“Growing tomatoes in a soilless substrate can provide a viable alternative to avoid the problems of soil salinization and soil-borne diseases caused by long-term continuous cropping in traditional cultivation system“.

Line 72, contents as well as?

Response: Changed “Wang et al. suggested that substrate-based cultivation could significantly increase the vitamin C contents well as increase the yield by 12.5%-16.8% [14]. “ to

“Wang et al. (2016) suggested that substrate-based cultivation could significantly increase the vitamin C, soluble solids and soluble sugar in tomato fruits, improve the sugar-acid ratio of taste quality and increase the yield by 12.5%-16.8% [14].”

Line 81-82 “?”.

Response: the spaces were removed”.

Line 98 to 102 and Table 1, So there are two different soilless substrates? Did they behave the same? What is the soil-based substrate? Where is it collected? What type of soil is it (description)?

Response: There are not two different soilless substrates. One is soilless substrates and another is soil. We change “Soil-based” to “Soil” in the body text and Table 1. The soil type in soil cultivation was loam, and the soil was collected and transferred from the experimental field into greenhouse at the Modern Agricultural Experiment Park of the Hebei Academy of Agriculture and Forestry, Hubei province in China.

Line 110-111 Table 2, I don't understand the water measurement. Water would be litres or maybe some volume (say cubic cms?) per day. This listed measurement is not a volume??

 Response: The “square meter/day” was changed to “1000 liters/day” in Table 2.

Line 113-125, "soon after harvest" does not allow for replication. Next day? next week? Were the SPAD meter readings replicated? If so, different leaves or same leaf with several readings? When were fruits harvested? Harvest time (and fruit maturity) play substantial roles in fruit quality.

Response:

(1) Changed “Plant height and stem thickness were measured in 50 days after planting; root volume, roots, stems, and leaves fresh weight, and relative chlorophyll content were measured soon after harvesting. Fifteen plants were evaluated for each trait with three replications. The chlorophyll content was measured with a 502-chlorophyll meter (Konica Mi- 115 nolta, Japan).” To

“Plant height, stem thickness, root volume, roots, stems, leaf fresh weight, and relative chlorophyll content were measured in 50 days after planting. Fifteen plants were evaluated for each trait with three replicates. The chlorophyll content was measured with a 502-chlorophyll SPAD meter (Konica Minolta, Japan). Each leaf was measured three times in same day; the average of the three readings was recorded; and 15 plants were measured for each treatment. “

The new sentence is “The mature fruits were harvested several times when the fruit color turns completely red. The total numbers of fruits were not recorded from each treatment but the yield was recorded cumulatively harvesting in 50-plants growing in a 17-meter long row and converted into yield per mu (1 mu = 667 m2).”

Line 135-136, What maturity stage? I assume you are using apparently fully ripe tomatoes, determined by color. You should say so. How many fruits were combined?

Response: Yes, we used the fully ripe tomatoes. The “during the fruit maturity stage” was changed to “with fully ripe tomatoes when the fruit color turns completely red”.

Line 136, explain this “mixed”.

Response: “, mixed and” was removed.

Line 213-214, need a reference.

Response: Two references were added as below:

Nitrate is an important indicator to measure the quality of vegetables with negative impact [23]. 70%-80% of the nitrate ingested by humans comes from vegetables. When it accumulates in the body to a certain level, nitrate will affect the human body by serious health hazards [24]. Excessive nitrate content in the tomato fruit is considered as a detrimental factor for human health [23;24].

  1. Yang, W.; Hu, X. Research progress of effect of nitrogen fertilizer on Vc and nitrate content in vegetables. Journal of An Hui Agricultural Sciences, 2006, 34(22), 5924-5925.
  2. Chen, Z.; Feng, D. Changes of nitrate and nitrite contents and their chemical regulation of leafy vegetables. Chinese Bulletin of Botany, 1994, 11(3), 25-26.

Line 216, reword this sentence, the "proper ratio" can produce fruits with good flavor (why say taste and flavor? How are those thing different?)

Response: We reword the sentence as “A proper sugar-acid ratio can make tomatoes have a good taste and flavor” was changed to “A proper sugar-acid ratio can produce fruits with good flavor”.

In conclusion, “I don't understand if the two soilless medium formulations acted the same or not? I don't have any idea what sorts of soil medium was used, it is NOT described. I'd be hesitant to agree with the claims until the soil system is explained better”.

Response: Sorry we did not describe clearly. There were not two soilless cultivation, but only one soilless medium was used. The another is a normal soil condition as control in the experiments.

Round 2

Reviewer 2 Report

Thank you for clarifying some areas of concern, improved the manuscript.